# Increase in Toxicity of Anticancer Drugs by PMTPV, a Claudin-1-Binding Peptide, Mediated via Down-Regulation of Claudin-1 in Human Lung Adenocarcinoma A549 Cells

**DOI:** 10.3390/ijms21165909

**Published:** 2020-08-17

**Authors:** Haruka Nasako, Yui Takashina, Hiroaki Eguchi, Ayaka Ito, Yoshinobu Ishikawa, Toshiyuki Matsunaga, Satoshi Endo, Akira Ikari

**Affiliations:** 1Laboratory of Biochemistry, Department of Biopharmaceutical Sciences, Gifu Pharmaceutical University, Gifu 501-1196, Japan; 155050@gifu-pu.ac.jp (H.N.); 145037@gifu-pu.ac.jp (Y.T.); 146008@gifu-pu.ac.jp (H.E.); 175012@gifu-pu.ac.jp (A.I.); sendo@gifu-pu.ac.jp (S.E.); 2Department of Physical Biochemistry, School of Pharmaceutical Sciences, University of Shizuoka, Shizuoka 422-8526, Japan; ishi206@u-shizuoka-ken.ac.jp; 3Education Center of Green Pharmaceutical Sciences, Gifu Pharmaceutical University, Gifu 502-8585, Japan; matsunagat@gifu-pu.ac.jp

**Keywords:** claudin-1, lung adenocarcinoma, chemoresistance, short peptide

## Abstract

Claudin-1 (CLDN1), a tight junctional protein, is highly expressed in lung cancer cells and may contribute to chemoresistance. A drug which decreases CLDN1 expression could be a chemosensitizer for enhancing the efficacy of anticancer drugs, but there is no such drug known. We found that PMTPV, a short peptide, which mimics the structure of second extracellular loop (ECL2) of CLDN1, can reduce the protein level of CLDN1 without affecting the mRNA level in A549 cells derived from human lung adenocarcinoma. The PMTPV-induced decrease in CLDN1 expression was inhibited by monodansylcadaverine, a clathrin-mediated endocytosis inhibitor, and chloroquine, a lysosome inhibitor. Quartz crystal microbalance assay showed that PMTPV can directly bind to the ECL2 of CLDN1. In transwell assay, PMTPV increased fluxes of Lucifer yellow (LY), a paracellular flux marker, and doxorubicin (DXR), an anthracycline anticancer drug, without affecting transepithelial electrical resistance. In three-dimensional spheroid culture, the size and cell viability were unchanged by short peptides, but the fluorescence intensity of hypoxia probe LOX-1 was decreased by PMTPV. PMTPV elevated the accumulation and cytotoxicity of DXR in the spheroids. Similar results were observed by knockdown of CLDN1. Furthermore, the sensitivities to cisplatin (CDDP), docetaxel, and gefitinib were enhanced by PMTPV. The level of CLDN1 expression in CDDP-resistant cells was higher than that in parental A549 cells, which was reduced by PMTPV. PMTPV restored the toxicity to DXR in the CDDP-resistant cells. Our data suggest that PMTPV may become a novel chemosensitizer for lung adenocarcinoma.

## 1. Introduction

Non-small cell lung cancer (NSCLC) accounts for 80% of all lung cancer cases and is commonly insensitive and intrinsically resistant to original chemotherapy [1]. Cisplatin (CDDP)-based combination chemotherapy regimens have been the standard therapeutic strategy in advanced stage NSCLC. The patients show the response rate to chemotherapy is about 50% in first-line treatments, but the patients acquire resistance to chemotherapy and the efficacy drops to about 15% in second- or third-line treatments [2,3]. Other traditional chemotherapeutic agents such as gemcitabine, etoposide, and doxorubicin are often used as combination therapy, but they also induce chemoresistance. These drugs can acquire cross-resistance to a wide variety of anticancer agents which have no obvious structural or functional similarities [4]. Resistance remains an obstacle in chemotherapy and seriously influences the survival rate of NSCLC patients.

Epithelial cells form three types of cell-cell junction named tight junctions (TJs), adherens junctions, and desmosomes. Among them, the TJs are located at the most apical section of the plasma membrane of adjacent cells. Claudins (CLDNs) are a major component of TJs and form a family of over 20 closely related transmembrane proteins [5]. The expression levels of some CLDN subtypes are high in cancer tissues such as lung, liver, breast, pancreatic, and bladder carcinomas [6,7]. Abnormal expression of CLDN subtypes enhances proliferation and migration in cancer cells, but the pathophysiological function of CLDNs is not fully understood. Epithelial mesenchymal transition (EMT) is proposed to have an important role in the acquisition of invasive and metastatic abilities of cancer cells. The reduction of CLDN3 and CLDN18 expression induces EMT in squamous cell carcinoma cells [8] and adenocarcinoma cells [9], respectively. The expression level of CLDN1 is increased in colon [10], stomach [11], and thyroid cancer tissues [12]. So far, we reported that the expression level of CLDN1 is increased by the acquisition of chemoresistance to CDDP in lung adenocarcinoma A549 cells [13].

Peptides are biologically active substances involved in regulating various physiological functions. They often show higher affinity and specificity, and have lower adverse effects compared with small-molecule drugs [14]. There are 60–70 approved peptide drugs in the US, Europe, and Japan, and 260 compounds are tested in clinical trials [15]. In recent years, up to 20 of the antitumor peptide drugs have entered into clinical. CIGB-300 (a cell-permeable selective CK2 inhibitory peptide), abarelix (a novel antagonist of gonadotropin-releasing hormone), and p28 (a non-human double minute 2-mediated peptide inhibitor of p53 ubiquitination) are clinical trials for cervical carcinoma [16], prostate cancer [17], and advanced solid tumors [18], respectively. CLDN-targeting peptides have been reported against the second extracellular loop (ECL2) of CLDNs. DFYNP, which mimics the structure of ECL2 of CLDN3 and CLDN4, reduces the protein levels of these CLDNs and induces apoptotic cell death in breast cancer cells [19]. We recently reported that both VPDSM and DSMKF, with structures that are not similar to the ECL2 of CLDN3 and CLDN4, decrease the protein level of CLDN2 and enhance chemosensitivity of in vitro spheroids of lung adenocarcinoma cells [20]. The elevation of CLDN1 expression in A549R cells enhances chemoresistance to anticancer drugs [13], but there is no therapeutic drug, which can reduce CLDN1 expression.

In the present study, we searched for a short peptide, which can selectively decrease the expression of CLDN1 in A549 cells. The mRNA and protein levels of CLDN1 were investigated by quantitative real-time polymerase chain reaction (PCR) and Western blotting, respectively. The cellular localization of CLDN1 was assessed by an immunofluorescence assay. Quartz crystal microbalance (QCM) analysis was performed to detect direct interaction between a short peptide and the ECL2 of CLDN1. Paracellular permeability and chemosensitivity to anticancer drugs were investigated using two-dimensional (2D) and 3D culture models, respectively. Chemosensitivity to anticancer drugs was estimated by the change of adenosine triphosphate content in the spheroids.

## 2. Results

### 2.1. Expression of CLDN1 in Lung Tissues

CLDN1 is constitutively expressed in human lung tissues [21]. The mRNA level of *CLDN1* was significantly increased in human lung adenocarcinoma and squamous cell carcinoma tissues compared to normal tissues, and has a tendency to increase in large cell and small cell tissues (Figure 1A). CLDN1 expression was also checked using lung cancer cell lines. The mRNA level of *CLDN1* in A549 cells derived from lung adenocarcinoma, RERF-LC-AI cells derived from squamous cell carcinoma, IA-LC cells derived from large cell carcinoma, and WA-hT cells derived from small cell carcinoma were higher than that in normal tissue (Figure 1B). Especially, CLDN1 was highly expressed in A549 cells.

### 2.2. Decrease in Protein Level of CLDN1 by Short Peptides

CLDN2, 3, and 4 have been reported to be decreased by short peptides, which mimic ECL2 of each CLDN, in lung adenocarcinoma and breast cancer cells [19,20,22]. The primary structure of ECL2 of CLDN1 differs from those of CLDN2, 3, and 4. We examined the effects of five types of short peptides (EFYDP, YDPMT, PMTPV, TPVNA, and VNARY), which mimic ECL2 of CLDN1, on CLDN1 expression. These peptides showed no significant toxicity in A549 cells (Figure 2B). The protein level of CLDN1 was decreased by PMTPV and TPVNA (Figure 2C). EFYDP, YDPMT, and VNARY slightly decreased that of CLDN1, but the effects were not significant. To clarify the mechanism of a short peptide, we investigated the effect of PMTPV, which shows the most significant decrease in CLDN1 expression, in detail. The protein level of CLDN1 was dose-dependently decreased by PMTPV, whereas that of CLDN2 was unchanged (Figure 2D). Neither the mRNA levels of *CLDN1* nor *CLDN2* were decreased by PMTPV (Figure 2E). These results indicate that transcriptional activity may not be involved in the PMTPV-induced decrease in CLDN1 expression.

### 2.3. Rescue of PMTPV-Induced Decrease in CLDN1 Expression by Endocytosis and Lysosome Inhibitors

So far, we reported that the VPDSM and DSMKF-induced decrease in CLDN2 expression is inhibited by chloroquine (CQ), a lysosome inhibitor, and monodansylcadaverine (MDC), a clathrin-mediated endocytosis inhibitor [20]. Similarly, the PMTPV-induced decrease in CLDN1 expression was significantly suppressed by CQ and MDC (Figure 3A). Immunofluorescence measurement showed that CLDN1 was colocalized with ZO-1 under control conditions (Figure 3B). The red fluorescence signal of CLDN1 disappeared by the treatment with PMTPV without affecting the green fluorescence signal of ZO-1. A dot of the fluorescence of CLDN1 is apparent in the cytoplasm of the cells treated with PMTPV plus CQ. In contrast, most of CLDN1 was colocalized with ZO-1 at the cell-cell junction area in the cells treated with PMTPV plus MDC, indicating that PMTPV may enhance the clathrin-mediated endocytosis of CLDN1.

### 2.4. Interaction of PMTPV With ELC2 of CLDN1

The Ch1 of sensor chip was coated with the ECL2 of CLDN1 fused with biotin, whereas the Ch2 was coated with biotin, a negative probe (Figure 4A). The subtracted value of frequency of both the signals was increased by the addition of PMTPV, but not by bovine serum albumin (BSA) (Figure 4B). These results indicate that PMTPV can directly bind to the ECL2 of CLDN1.

### 2.5. Effects of PMTPV and CLDN1 siRNA on Paracellular Permeability

Paracellular permeability was assessed using cells cultured on transwell inserts. TER was not changed by PMTPV (Figure 5A). We recently reported that TER is not changed by CLDN1 overexpression [13]. These data indicate that CLDN1 may not function as an ion permeable channel. On the contrary, the paracellular fluxes of LY and DXR were increased by PMTPV (Figure 5B). Therefore, we hypothesized that CLDN1 may form a barrier to small molecules. To support the hypothesis, we investigated the effect of CLDN1 siRNA on paracellular permeability. The protein level of CLDN1 was significantly decreased by CLDN1 siRNA (Figure 5C). CLDN1 siRNA increased the paracellular fluxes of LY and DXR without affecting TER (Figure 5D,E).

### 2.6. Effects of Peptides on the Characteristics of 3D Culture Model

The size and viability of spheroid cells were not significantly changed by the treatments with five types of peptides for 24 h (Figure 6A–C). The fluorescence intensity of hypoxia probe LOX-1 was slightly but significantly decreased by PMTPV (Figure 6A,D). In contrast, other peptides had no effect on hypoxia level of spheroids.

### 2.7. Rescue of Toxicity to Anticancer Drugs by PMTPV and CLDN1 siRNA in Spheroids

The accumulation and toxicity of DXR in spheroid cells were increased in a dose-dependent manner (Figure 7). These effects were enhanced by PMTPV. The sensitivities to CDDP, GEF, and DOC were also enhanced by PMTPV. Instead of PMTPV, the effect of CLDN1 siRNA on the accumulation and toxicity of DXR was examined. The spheroid size was not significantly changed by CLDN1 siRNA, but the accumulation and toxicity of DXR were enhanced (Figure 8). These results are similar to those in PMTPV.

### 2.8. Rescue of Toxicity to Anticancer Drugs by PMTPV in the CDDP-Resistant Cells

So far, we reported that the toxicity of DXR was increased by CLDN1 siRNA in CDDP-resistant A549 (A549R) cells [13]. The protein level of CLDN1 in A549R cells was higher than that in parental cells, which was suppressed by PMTPV (Figure 9A). The accumulation and toxicity of DXR were enhanced by PMTPV in the spheroid of A549R cells (Figure 9B,C). These results are similar to those in CLDN1 siRNA [13].

## 3. Discussion

CLDN1 is necessary to regulate physiological barrier function in various tissues, but its pathophysiological role in cancer is controversial [23]. CLDN1 up-regulates invasive activity in the colorectal cancer cells [24]. In contrast, the survival rate of patients with attenuated CLDN1 expression is lower than that with preserved CLDN1 expression, and lower CLDN1 induces the invasion of hepatocellular carcinoma [25]. The expression of CLDN1 is increased in the A549R cells, leading to the reduction of accumulation and chemosensitivity of anticancer drugs in the spheroids [10]. In the present study, we found that the mRNA level of *CLDN1* in the human tissues of NSCLC is significantly higher than that in normal tissue (Figure 1). In addition, *CLDN1* was highly expressed in cell lines derived from human lung cancer tissues. CLDN1 correlates with poor prognosis in lung adenocarcinoma [26]. Therefore, CLDN1 may be involved in the malignant progression of NSCLC.

Traditional chemotherapeutic agents such as CDDP, DXR, and gemcitabine induce chemoresistance in various solid tumors including NSCLC [27,28]. Some patients treated with these drugs can develop resistance within several years and acquire cross-resistance. Epidermal growth factor receptor (EGFR)-tyrosine kinase inhibitors (TKIs) are recently being used for the first-line treatment of advanced NSCLC and activating EGFR mutations [29]. The EGFR-TKIs show notable effects in the patients with EGFR-activating mutations, but the rate of EGFR-activating mutation is only 11–14% in USA, 27–47% in Japan, 7% in Australia, and 24% in China [30], and some of them eventually relapse within one year from the onset of treatment [31]. Therefore, the development of chemosensitizer must be necessary to improve the antitumor properties. So far, we found that the knockdown of CLDN1 expression elevates chemosensitivity of A549 cells to anticancer agents [13]. Therefore, a drug which can reduce CLDN1 expression may be a novel chemosensitizer for lung adenocarcinoma. In addition, CLDN1 was highly expressed in not only other NSCLC types but also colon [10], stomach [11], and thyroid cancer tissues [12], suggesting that PMTPV may enhance the chemosensitivity of these cancer cells. CLDN1 is also expressed in normal lung tissues [21], but its physiological role has not been fully understood. In future study, it is necessary to investigate the adverse effect of PMTPV on lung function.

PMTPV decreased CLDN1 expression in a dose-dependent manner without affecting CLDN2 expression (Figure 2D), suggesting that the peptide selectively acts on CLDN1. Furthermore, PMTPV decreased neither mRNA levels of *CLDN1* nor *CLDN2* (Figure 2E), suggesting that transcriptional regulation is not involved in the reduction of *CLDN1* by PMTPV. The idea is supported by the data that the PMTPV-induced decrease in CLDN1 expression was rescued by the clathrin-dependent endocytosis inhibitor and lysosome inhibitor (Figure 3). QCM assay showed that PMTPV is directly associated with ECL2 of CLDN1 (Figure 4). We recently reported that both VPDSM and DSMKF decrease the protein level of CLDN2 without affecting that of CLDN1 in A549 cells [20]. These short peptides may be selectively bound to each CLDN subtype.

The molecular mechanism of chemoresistance is multifactorial and poorly understood. We found that CLDN1 functions as a barrier against low-molecule compounds (Figure 5). PMTPV and CLDN1 siRNA increased the accumulation and toxicity of DXR in the spheroids (Figure 7 and Figure 8). The chemoresistance may be reversed by the disruption of barrier function against anticancer agents in the spheroids. The acquisition of chemoresistance occurs due to gene mutation and aberrant elevation of target protein, target modification, drug inactivation, induction of apoptosis-resistant gene product, and elevation of drug efflux transporters [32]. Hypoxia is one of the most important factors that contribute to aggressiveness and chemoresistance of tumors. Hypoxia inducible factor-1 (HIF-1) is ubiquitinated and degraded via proteasome pathway under normoxic conditions, but it is stabilized and activated under hypoxic conditions. HIF-1 is highly expressed in 3D in vitro spheroid cultures of cancer cells [33]. The hypoxic level of A549 spheroid cells is increased by the overexpression of CLDN1 [13] and decreased by the knockdown of CLDN1 (Figure 6C). The hypoxic level may be also involved in the improvement of chemoresistance by PMTPV. CLDN1 has been reported to increase drug resistance of NSCLC by activating autophagy in 2D culture models [34]. Autophagy also contributes to the chemoresistance of NSCLC [35]. We need further study to clarify the involvement of autophagy.

A carboxyl-terminal fragment of the bacterial *Clostridium perfringens* enterotoxin (cCPE) can bind to some CLDNs including CLDN3, 4, 9, 15, and 19, and disrupt epithelial barriers [36]. Crystal structures of these CLDNs have recently been solved [37,38,39,40,41]. Mutation and binding analyses reveal that an ECL2 motif with sequence NP(V/L)(V/L)(*p*/A153) of CLDNs is required for the interaction between CLDN and cCPE. However, it is unknown whether cCPE prevents the assembly of CLDNs before forming complete TJs and breaks cis or trans interactions of CLDNs. QCM assay showed that PMTPV binds to the ECL2 of CLDN1 (Figure 4). We suggest that PMTPV at least penetrates into the cis-interaction space and breaks the interaction. Further studies are needed to clarify the binding characteristics of PMTPV with CLDN1.

In conclusion, we found that PMTPV, which mimics the ECL2 of CLDN1, decreases the protein level of CLDN1 in A549 cells without affecting mRNA level of *CLDN1*. The PMTPV-induced decrease in CLDN1 expression was inhibited by MDC and CQ, suggesting that PMTPV enhances the endocytosis of CLDN1 from the TJs and degradation in the lysosome. PMTPV increased paracellular permeabilities to LY and DXR in 2D culture models, and enhanced chemosensitivity to anticancer drugs in 3D culture models. These results coincided with those of CLDN1 siRNA. In addition, PMTPV could enhance chemosensitivity to DXR in the A549R cells. PMTPV may be useful to mitigate chemoresistance of NSCLC.4.

## 4. Materials and Methods

### 4.1. Materials

Goat anti-β-actin polyclonal antibody and gefitinib (GEF) were obtained from Santa Cruz Biotechnology (Santa Cruz, CA, USA). Chloroquine (CQ), CDDP, and DXR were from Fujifilm Wako Pure Chemical (Osaka, Japan). Rabbit anti-CLDN1 polyclonal antibody and mouse anti-ZO-1 antibody were from Thermo Fisher Scientific (Rockford, IL, USA). Docetaxel (DOC), lactacystin, Lucifer yellow (LY), and monodansylcadaverine (MDC) were from Tokyo Chemical Industry (Tokyo, Japan), Cayman Chemical (Ann Arbor, MI, USA), Biotium (Fremont, CA, USA), and Sigma-Aldrich (Saint Louis, MO, USA), respectively. Short peptides (EFYDP, YDPMT, PMTPV, TPVNA, and VNARY) were synthesized by GenScript (Piscataway, NJ, USA) and were dissolved in 30% dimethyl sulfoxide solution. All other reagents were of the highest purity available.

### 4.2. Cell Culture and Transfection

A549 cells derived from human lung adenocarcinoma were purchased from the RIKEN BRC through the National Bio-Resource Project of the MEXT (Ibaraki, Japan). The cells were cultured in Dulbecco’s modified Eagle’s medium (Fujifilm Wako Pure Chemical) supplemented with 5% fetal calf serum (HyClone, Logan, UT, USA), 0.07 mg/mL penicillin-G potassium, and 0.14 mg/mL streptomycin sulfate in a 5% CO2 atmosphere at 37 °C as described previously [17]. The CDDP-resistant cells were prepared as described previously [10]. In 2D cultures, cell viability was assessed using a Premix WST-1 Cell Proliferation Assay System (Clontech Takara-Bio, Tokyo, Japan). Small interfering RNA (siRNA) for CLDN1 was transfected into the cells using a ScreenFect A (Fujifilm Wako Pure Chemical).

### 4.3. Western Blotting

Cells cultured on 6 cm-dishes were scraped into cold phosphate-buffered saline. Western blotting was performed as described previously [10]. In brief, cell lysates were applied to sodium dodecyl sulfate-polyacrylamide gel electrophoresis and blotted onto a poly(vinylidene fluoride) membrane. The membrane was incubated with each primary antibody (1:1000 dilution) at 4 °C for 16 h, followed by a peroxidase-conjugated secondary antibody (1:3000 dilution) at room temperature for 1 h. Finally, the blots were incubated by EzWestLumi plus (Atto Corporation, Tokyo, Japan) and scanned using a C-DiGit Blot Scanner (LI-COR Biotechnology, Lincoln, NE, USA). Band density was quantified using ImageJ software (National Institute of Health, Bethesda, MD, USA).

### 4.4. Quantitative Reverse Transcription-PCR

Lung cancer cDNA arrays II and V, which contain 12, 45, 26, 4, and 3 samples of normal, adenocarcinoma, squamous cell carcinoma, large cell, and small cell, respectively, were purchased from OriGene (Rockville, MD, USA). Total RNA was isolated from the cells using TRI reagent (Molecular Research Center, Cincinnati, OH, USA). Reverse transcription and quantitative real-time PCR were performed as described previously [10].

### 4.5. Immunofluorescence Measurement

The cellular localization of CLDN1 was investigated by immunofluorescence measurement. Cells cultured on cover glasses were fixed with methanol for 10 min at −20 °C, permeabilized with 0.2% Triton X-100 for 15 min, and blocked with 4% Block Ace (Dainippon Sumitomo Pharma, Osaka, Japan) for 30 min. The cells were incubated with anti-CLDN1 and anti-ZO-1 antibodies for 16 h at 4 °C, and then incubated with DyLight 488- and 549-conjugated antibodies (Bio-Rad Laboratories, Hercules, CA, USA) in the presence of a nuclear marker 4′,6-diamidino-2-phenylindole (DAPI) for 1.5 h at room temperature. Images of stained cells were obtained using an LSM 700 confocal microscope (Carl Zeiss, Jena, Germany).

### 4.6. QCM Analysis

Recombinant protein of ECL2 of CLDN1 fused with biotin was prepared by GenScript. Ch1 of a QCM twin sensor chip (Nihon Dempa Kogyo, Tokyo, Japan) was incubated with recombinant CLDN1, whereas Ch2 was incubated with biotin. Short peptides (5 μg/mL) were injected into the flow cell at a flow rate of 50 μL/min. Detection of the association between short peptides and CLDN1 was carried out using a QCM sensor system. Sensor response was measured by subtracting the frequency shifts of Ch1 from Ch2.

### 4.7. Paracellular Flux Assay

Cells were cultured on transwell plates (0.4 μm pore size, 12 mm diameter, Corning Incorporated, Corning, NY, USA). The paracellular fluxes of ions and LY were measured by a Millicell-ERS volt ohm meter (Millipore, Temecular, CA, USA) and microplate reader F200 PRO (Tecan, Männedorf, Switzerland), respectively, as described previously [10,18].

### 4.8. 3D Spheroid Model

Cells were cultured on PrimeSurface96U multi-well plates (Sumitomo Bakelite, Tokyo, Japan). The size, viability, hypoxia level, and DXR accumulation were measured as described previously [18]. Chemosensitivity of spheroid was investigated by the cells treated with anticancer drugs including DXR, CDDP, GEF, and DOC for 24 h.

### 4.9. Statistical Analysis

Results are presented as means ± S.E.M. Differences between groups were analyzed by one-way analysis of variance, and corrections for multiple comparisons were made using Tukey’s or Dunnett’s test for multiple comparison. Comparisons between two groups were made using Student’s *t* test. Statistical analyses were performed using KaleidaGraph version 4.5.1 software (Synergy Software, Reading, PA, USA). Significant differences were assumed at *p* < 0.05.

## Figures and Tables

**Figure 1 ijms-21-05909-f001:**
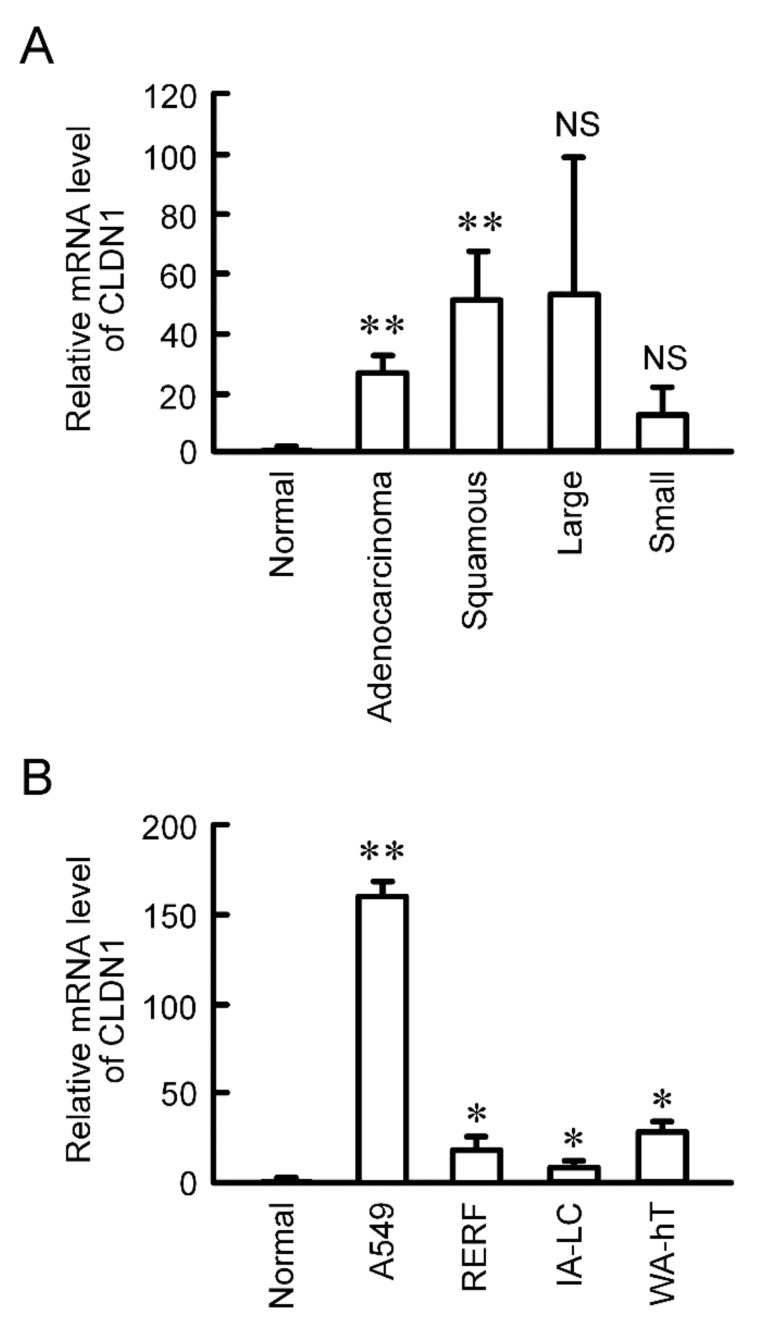
Elevation of *CLDN1* mRNA in human lung cancer tissues and cell lines. (**A**) cDNA was prepared using lung cancer cDNA arrays II and V, and then quantitative real-time PCR was performed using primer pairs for CLDN1 and β-actin. (**B**) cDNA was prepared from the total RNA extracted from A549, RERF-LC-AI (RERF), IA-LC, and WA-hT cells, and then real time-PCR was performed using primer pairs for CLDN1 and β-actin. The mRNA level of *CLDN1* is shown relative to the values in normal lung tissues (Normal). ** *p* < 0.01 and * *p* < 0.05 compared with normal. NS, *p* > 0.05.

**Figure 2 ijms-21-05909-f002:**
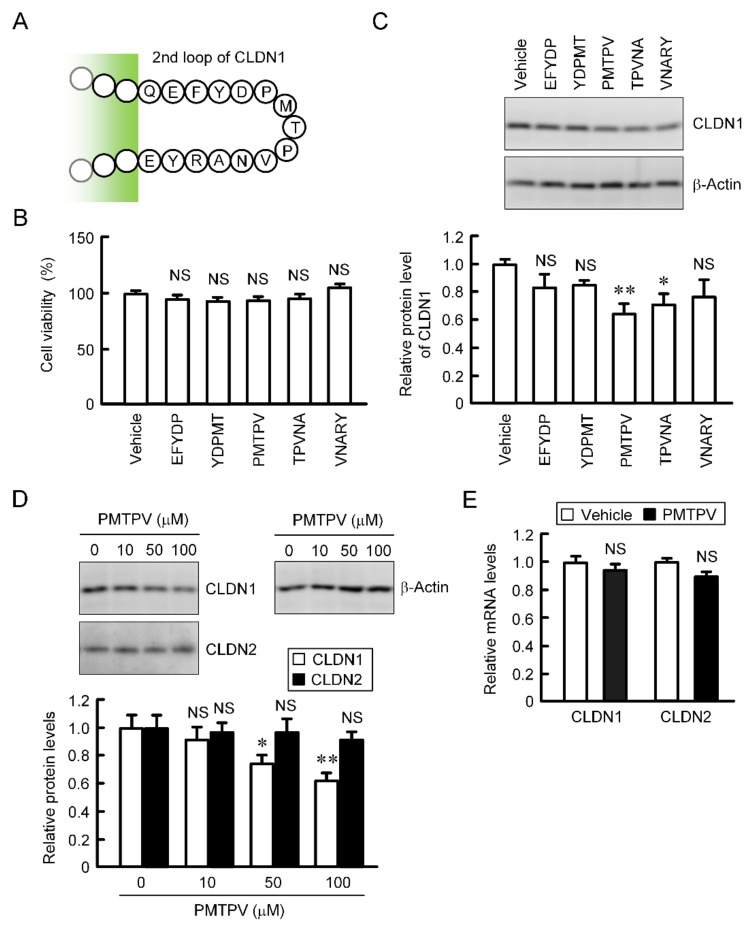
Effects of short peptides on the expression of CLDN1 in A549 cells. (**A**) Sequence of ECL2 of CLDN1. (**B**) A549 cells were incubated with each peptide at 100 μM or without peptide (vehicle) for 24 h. Cell viability was assessed by WST-1 assay. (**C**) The cell lysates were immunoblotted with anti-CLDN1 and anti-β-actin antibodies. The protein level of CLDN1 is represented relative to the values in vehicle. (**D**) Cells were incubated with PMTPV for 24 h at the concentration indicated. The protein levels of CLDN1 and CLDN2 are represented relative to the values in 0 μM. (**E**) Cells were incubated with PMTPV at 100 μM for 6 h. After extraction of total RNA, quantitative real-time PCR was performed using primer pairs for CLDN1, CLDN2, and β-actin. The mRNA levels of *CLDN1* (open bars) and *CLDN2* (closed bars) are represented relative to the values in vehicle. *n* = 3–4. ** *p* < 0.01 and * *p* < 0.05 compared with vehicle or 0 μM. NS, *p* > 0.05.

**Figure 3 ijms-21-05909-f003:**
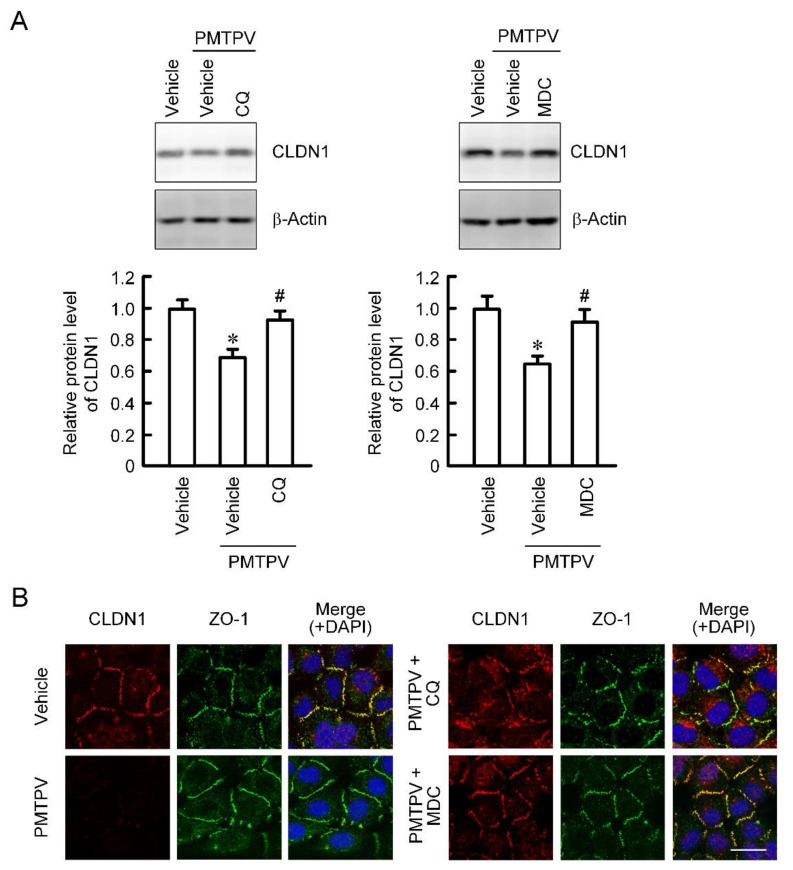
Inhibition of PMTPV-induced decrease in CLDN1 expression by endocytosis and lysosome inhibitors. A549 cells were incubated with PMTPV at 100 μM for 24 h in the presence and absence (vehicle) of 10 μM chloroquine (CQ) or 5 μM monodansylcadaverine (MDC). (**A**) The cell lysates were immunoblotted with anti-CLDN1 and anti-β-actin antibodies. The protein level of CLDN1 is represented relative to the values in vehicle. (**B**) The cells were stained with anti-CLDN1 (red), anti-ZO-1 (green) antibodies. Merged images with 4′,6-diamidino-2-phenylindole (DAPI) (blue) are indicated on the right. Scale bar represents 10 µm. *n* = 3–4. * *p* < 0.05 compared with vehicle. # *p* < 0.05 compared with PMTPV alone.

**Figure 4 ijms-21-05909-f004:**
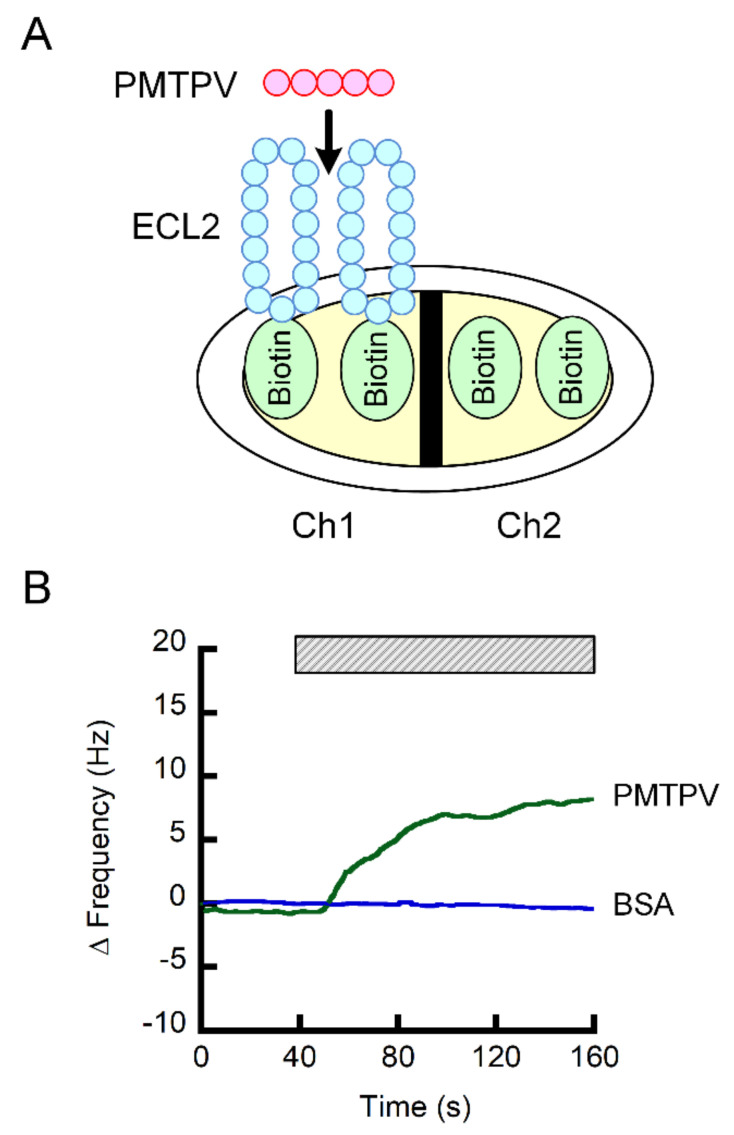
Interaction of PMTPV with ECL2 of CLDN1. (**A**) Scheme for the coating of biotin and biotin-fused ECL2 of CLDN1 on the sensor chip. (**B**) The QCM frequency was measured every 1 s. Then, 5 ng/mL PMTPV was applied at the time period indicated by the hatched box. Bovine serum albumin (BSA) was used as a non-specific protein for a negative control.

**Figure 5 ijms-21-05909-f005:**
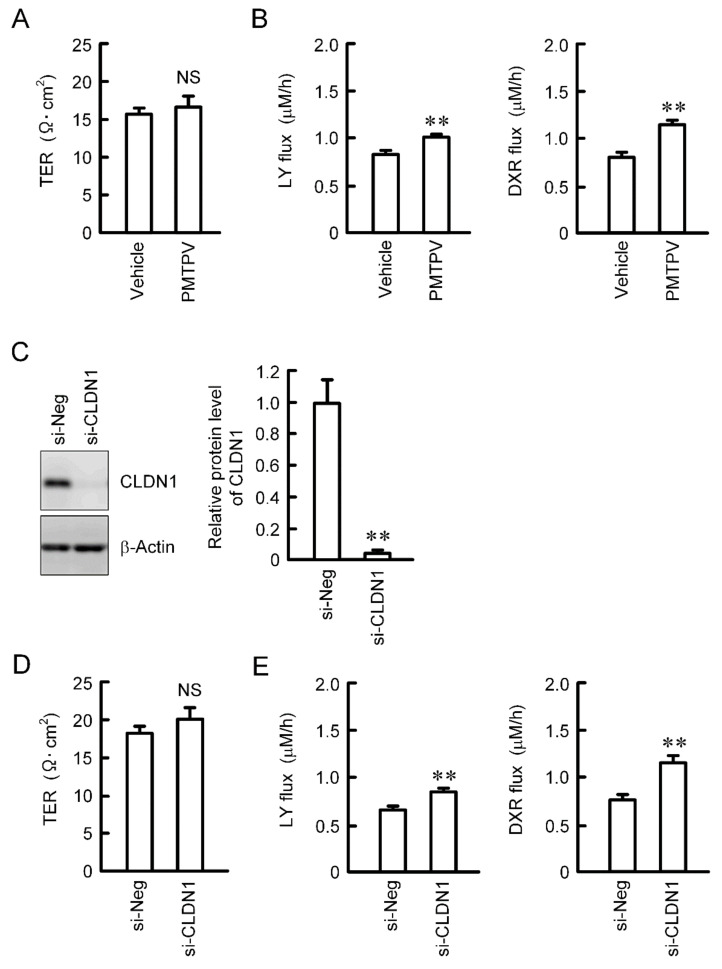
Increase in paracellular permeability of small molecules by PMTPV and CLDN1 siRNA. (**A**,**B**) A549 cells cultured on transwell plates were incubated in the presence and absence (vehicle) of 100 μM PMTPV for 24 h. TER was measured using a volt ohmmeter. LY (10 μM) or DXR (10 µM) was added to the apical compartment. After incubation at 4 °C for 1 h, the solution in the basal compartment was collected. The concentration of DXR and LY was measured using an Infinite F200 microplate reader. (**C**–**E**) A549 cells were transfected with negative (si-Neg) or CLDN1 siRNA (si-CLDN1). The protein level of CLDN1 was examined by Western blotting and represented relative to the values in the cells transfected with negative siRNA. Paracellular permeability was estimated by TER, LY flux, and DXR flux. *n* = 4–6. ** *p* < 0.01 compared with vehicle or si-Neg. NS, *p* > 0.05.

**Figure 6 ijms-21-05909-f006:**
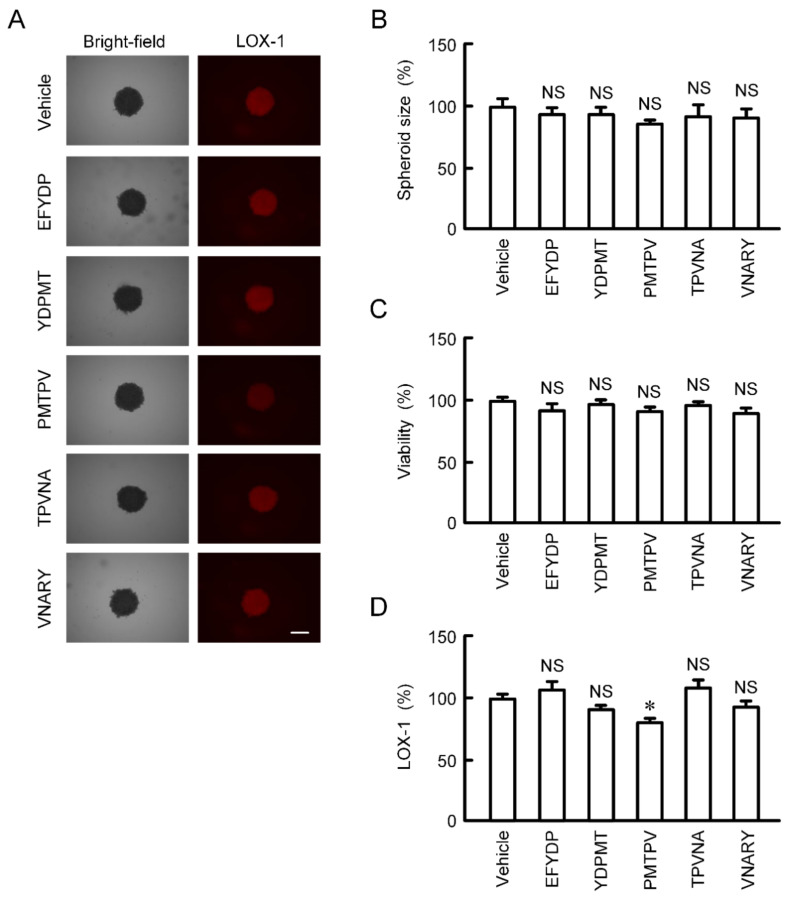
Effects of short peptides on cell viability and hypoxic level in 3D culture. A549 cells cultured on PrimeSurface96U multi-well plates were incubated in the presence and absence (vehicle) of each peptide at 100 μM for 24 h. (**A**) The images of spheroids were collected using a BZ-X800 fluorescence microscope. Representative images of bright-field and LOX-1 are shown. Scale bar indicates 500 μm. (**B**) The size was calculated using ImageJ and represented relative to the values in vehicle. (**C**) Cell viability was assessed using a CellTiter-Glo 3D Cell Viability Assay kit and represented relative to the values in vehicle. (**D**) Hypoxic level in spheroids was assessed using LOX-1 and represented relative to the values in vehicle. *n* = 4–6. * *p* < 0.05 compared with vehicle. NS, *p* > 0.05.

**Figure 7 ijms-21-05909-f007:**
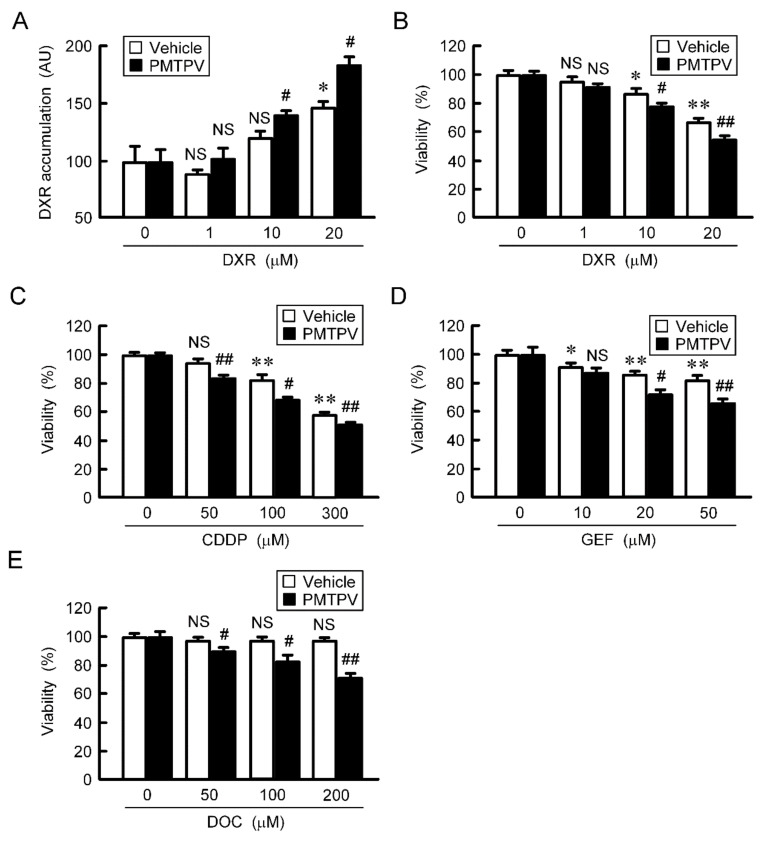
Reduction of chemoresistance by PMTPV in 3D culture. (**A**,**B**) A549 cells cultured on PrimeSurface96U multi-well plates were incubated in the presence and absence (vehicle) of 100 μM PMTPV for 24 h, and then they were incubated with DXR for 60 min (**A**) and 24 h (**B**) at the concentration indicated. The fluorescence intensity of DXR was measured using a fluorescence microscope and represented as arbitrary units (AU). Cell viability was assessed using a CellTiter-Glo 3D Cell Viability Assay kit and represented relative to the values in 0 µM DXR. (**C**–**E**) The cells were incubated with CDDP, GEF, or DOC in the presence and absence (vehicle) of PMTPV for 24 h. Cell viability was represented relative to the values in 0 µM anticancer drugs. *n* = 4–6. ** *p* < 0.01 and * *p* < 0.05 compared with 0 μM. ## *p* < 0.01 and # *p* < 0.05 compared with vehicle. NS, *p* > 0.05.

**Figure 8 ijms-21-05909-f008:**
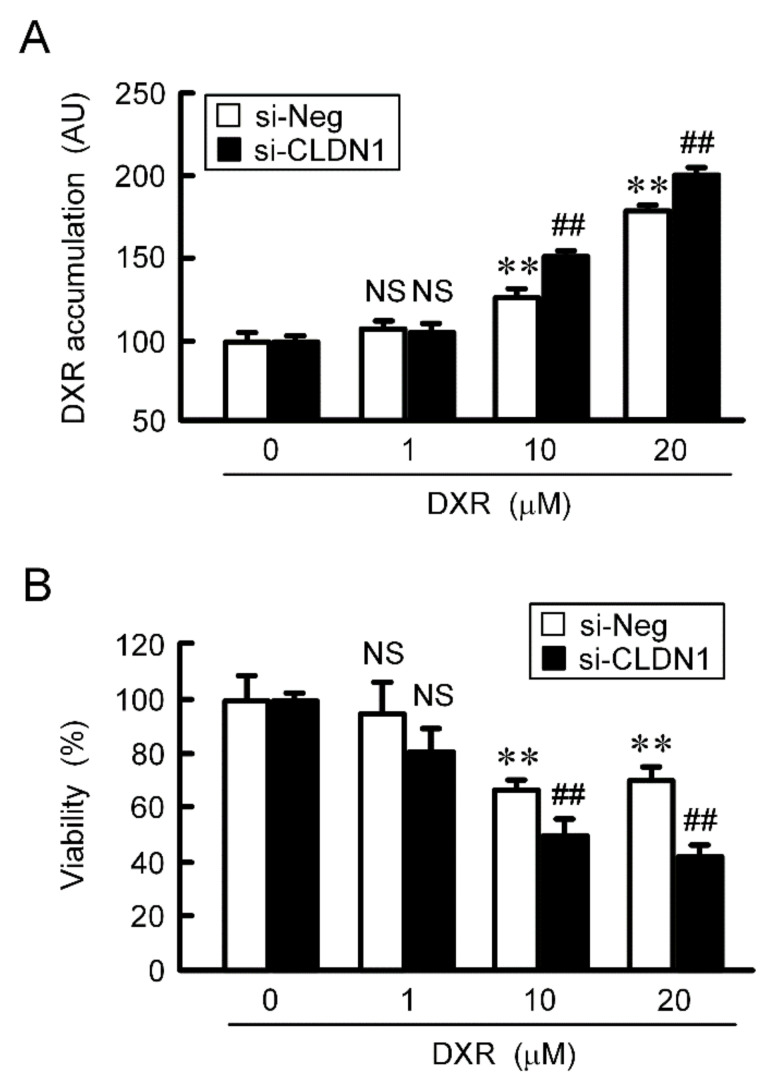
Reduction of chemoresistance by CLDN1 siRNA in 3D culture. A549 cells cultured on PrimeSurface96U multi-well plates were transfected with negative (si-Neg) or CLDN1 siRNA (si-CLDN1). Then, the cells were incubated in the presence and absence of DXR for 60 min (**A**) or 24 h (**B**) at the concentration indicated. The fluorescence intensity of DXR is represented as arbitrary units (AU). Cell viability is represented relative to the values in 0 µM DXR. *n* = 4–6. ** *p* < 0.01 compared with 0 μM. ## *p* < 0.01 compared with si-Neg. NS, *p* > 0.05.

**Figure 9 ijms-21-05909-f009:**
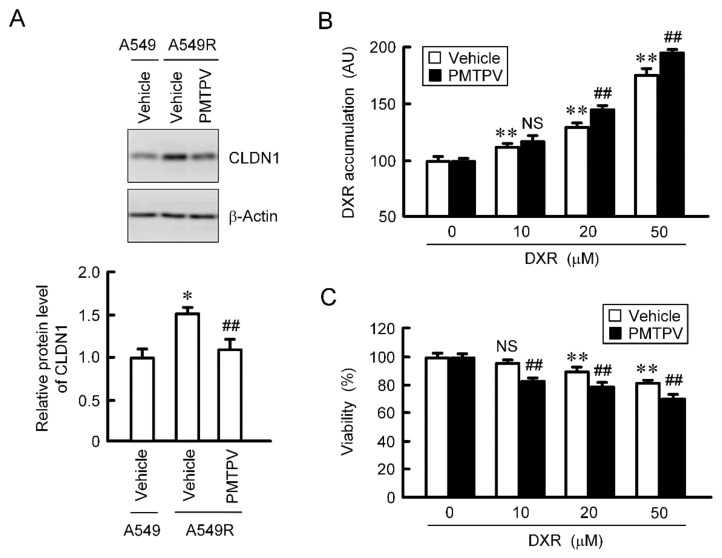
Reduction of chemoresistance by PMTPV in 3D culture of DXR-resistant cells. (**A**) A549 and A549R cells were incubated in the presence and absence (vehicle) of 100 μM PMTPV for 24 h. The cell lysates were immunoblotted with anti-CLDN1 and anti-β-actin antibodies. The protein level of CLDN1 is represented relative to the values in A549. * *p* < 0.01 compared with A549. ## *p* < 0.01 compared with vehicle of A549R. (**B**,**C**) A549R cells cultured on PrimeSurface96U multi-well plates were incubated with PMTPV for 24 h, and then incubated with DXR for 60 min (**B**) or 24 h (**C**) at the concentration indicated. The fluorescence intensity of DXR is represented as AU. Cell viability is represented relative to the values in 0 µM DXR. *n* = 4–6. ** *p* < 0.01 compared with 0 μM. ## *p* < 0.01 compared with vehicle. NS, *p* > 0.05.

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
