# Peer review of "Increase in Toxicity of Anticancer Drugs by PMTPV, a Claudin-1-Binding Peptide, Mediated via Down-Regulation of Claudin-1 in Human Lung Adenocarcinoma A549 Cells"

_ijms, 2020, doi:10.3390/ijms21165909_

Round 1
Reviewer 1 Report
The authors investigated PMTPV, an analogue of the ECL2 component of CLDN1, known to be involved in chemoresistance in cancer. They were able to demonstrate how PMTPV increased cell membrane paracellular flux without affecting transepithelial electrical resistance. With the use of three-dimensional spheroid culture, the authors suggested that PMTPV elevated the accumulation and cytotoxicity of DXR in these spheroids, similar to what seen with CLDN1 knockdown. As per their original hypothesis, the findings of the study seem to suggest that sensitivity to cisplatin (CDDP), docetaxel, and gefitinib were enhanced by PMTPV, but also that PMTPV restored the toxicity to DXR in previously CDDP-resistant cell A549 lung adenocarcinoma cells. Comments to the authors are as follows.
- What the potential effects of non-specific delivery of MPTPV in non-cancerous cells? The authors should elaborate a little more on this.
- As the A549 cell line derives from lung adenocarcinoma, it would be interesting to see if similar results are seen in other NSNCL types, and especially squamous cell carcinoma. This should be mentioned in the discussion.
Author Response
We thank you very much for your careful reading of our manuscript and valuable comments.
Comment 1
What the potential effects of non-specific delivery of PMTPV in non-cancerous cells? The authors should elaborate a little more on this.
Answer
Following your suggestion, we discussed the nonspecific effect of PMTPV on non-cancerous cells. Please see line 251.
Comment 2
As the A549 cell line derives from lung adenocarcinoma, it would be interesting to see if similar results are seen in other NSCLC types, and especially squamous cell carcinoma. This should be mentioned in the discussion.
Answer
Following your suggestion, we described the effect of PMTPV on other NSCLC types in the Discussion. Please see line 248.
Reviewer 2 Report
The authors investigated the role of short peptide PMTPV in increasing the activity of anticancer drugs against non-small cell lung cancer. This small peptide was able to inhibit Claudin-1 expression at the protein level, which was associated with a decrease in drug resistance. The authors conducted a comprehensive study to explain the behaviour of PMTPV against CLDN proteins on 2D and 3D models. Generally speaking, the topic of the article is quite innovative, due to the fact that there are currently no substances that could suppress claudins expression and support the applied chemotherapy. In my opinion, the presented article should interest IJMS readers. But there are certain points need to be addressed:
- the introduction is too brief, for example the authors can describe the mechanism of resistance and the role of claudins in the epithelial-mesenchymal transition, a process
that favors the spread of carcinomas, generation of cancer stem cells or tumor-initiating cells
- authors can add clinical trial numbers so that interested readers can quickly find them
- the authors must correct the symbols: beta, micro, degree. Part of the notation is correct, but most of the manuscript has "@" type errors
- "in vitro" - italics (line 68 and 264)
- line 78, the dot at the end of sentence
- authors must correct the all of references
- figure 1 authors may additionally title the graphs as: tissue/cell line
- why did not the authors decide to take a closer look at the mechanism of TPVNA on CLDN?
- why did the authors use two incubation times of short peptide with cells for protein and transcript levels investigation? Perhaps a comparable time could show similar effects on both molecular levels
- figure 6, I think the authors could consider adding spheroids images to the figure after PMTPV treatment and LOX-1 staining
- the authors mention the effect on hypoxia, what is the direct change in the hif-1 level (the main protein regulating this process)? What is the influence of the examined peptide on the process of autophagy, which can play a large role in resistance to treatment? Can a change in the level of claudins significantly inhibit autophagy?
- line 288, 4. materials and methods delete
- line 374 numbering: instead of 2.8.3. it is 2.8. 3D spheroids model
Author Response
We thank you very much for your careful reading of our manuscript and valuable comments.
Comment 1
The introduction is too brief, for example the authors can describe the mechanism of resistance and the role of claudins in the epithelial-mesenchymal transition, a process that favors the spread of carcinomas, generation of cancer stem cells or tumor-initiating cells
Answer
Following your suggestion, we described the correlation between EMT and claudin expression in the Introduction. Please see line 53.
Comment 2
Authors can add clinical trial numbers so that interested readers can quickly find them
Answer
The data are summarized in the review article. Therefore, we cited the article. Please see line 40.
Comment 3
The authors must correct the symbols: beta, micro, degree. Part of the notation is correct, but most of the manuscript has "@" type errors
Answer
The mistakes were made by publisher. Following your suggestion, we corrected the errors of symbols.
Comment 4
"in vitro" - italics (line 68 and 264)
Answer
Thank you for your pointed out. The mistakes were made by publisher. We corrected them.
Comment 5
Line 78, the dot at the end of sentence
Answer
Thank you for your pointed out. We corrected it.
Comment 6
Authors must correct the all of references
Answer
Thank you for your pointed out. The mistakes were made by publisher. We corrected them.
Comment 7
Figure 1 authors may additionally title the graphs as: tissue/cell line
Answer
Thank you for your suggestion. We modified them.
Comment 8
Why did not the authors decide to take a closer look at the mechanism of TPVNA on CLDN?
Answer
As shown in figure 2C, PMTPV shows most significant decrease in CLDN1 expression. Therefore, we decided to examine the mechanism of PMTPV. Please see line 107.
Comment 9
Why did the authors use two incubation times of short peptide with cells for protein and transcript levels investigation? Perhaps a comparable time could show similar effects on both molecular levels
Answer
The decrease in mRNA level occurs prior to the decrease in protein. Therefore, the effect of short peptide on the mRNA levels of CLDNs was examined at 6 h. We investigated the effect of 24-h treatment of PMTPV on the mRNA levels of CLDN1 and CLDN2 just in case. However, the mRNA levels were not significantly changed (data not shown).
Comment 10
Figure 6, I think the authors could consider adding spheroids images to the figure after PMTPV treatment and LOX-1 staining
Answer
Following your suggestion, we show the images of spheroids. Please see new figure 6.
Comment 11
The authors mention the effect on hypoxia, what is the direct change in the hif-1 level (the main protein regulating this process)?
Answer
HIF-1 is ubiquitinated and degraded via proteasome pathway under normoxic conditions, but it is stabilized and activated under hypoxic conditions. Please see line 270.
Comment 12
What is the influence of the examined peptide on the process of autophagy, which can play a large role in resistance to treatment? Can a change in the level of claudins significantly inhibit autophagy?
Answer
CLDN1 has been reported to increase drug resistance of NSCLC by activating autophagy in 2D culture model (#1). Autophagy also contributes to the chemoresistance of NSCLC (#2). At present, we do not know whether PMTPV can change autophagy.
#1 Zhao, Z.; Li, J.; Jiang, Y.; Xu, W.; Li, X.; Jing, W. CLDN1 Increases drug resistance of non-small cell lung cancer by activating autophagy via up-regulation of ULK1 phosphorylation. Med. Sci. Monit. 2017, 23, 2906-2916.
#2 Lee, J.G.; Shin, J.H.; Shim, H.S.; Lee, C.Y.; Kim, D.J.; Kim, Y.S.; Chung, K.Y. Autophagy contributes to the chemo-resistance of non-small cell lung cancer in hypoxic conditions. Respir. Res. 2015, 16, 138.
Comment 13
Line 288, 4. materials and methods delete
Answer
We think that it is needed to separate the sections of “discussion” and “materials and methods”.
Comment 14
Line 374 numbering: instead of 2.8.3. it is 2.8. 3D spheroids model
Answer
Thank you for your pointed out. The mistake was made by publisher. We corrected it.